# Effects of Temperature and Moisture Levels on Vitamin A in Total Mixed Ration Silage

**Pengjiao Tian, Huiying Hu, Xiya Zhang, Mingqing Chen and Xiqing Wang \***

College of Food Science Technology and Chemical Engineering, Hubei University of Arts and Science, Xiangyang 441053, China; tianpj0627@126.com (P.T.); huhuiying2023@163.com (H.H.); zhangxiya1113@163.com (X.Z.); m19902031871@163.com (M.C.)

\* Correspondence: xiqingwang91@163.com

**Abstract:** The aim of this study was to evaluate the effects of temperature and moisture levels on vitamin A in the total mixed ration (TMR) silage. The moisture levels of TMR were adjusted to 450 g/kg, 525 g/kg and 600 g/kg. Each moisture level had three replications and fermented at 15 °C, 30 °C and 40 °C, respectively. The TMR was sampled after 0, 7, 14, 28 and 56 days of ensiling. The fermentation quality, chemical composition and vitamin A content were analyzed. Correlations between vitamin A and individual fermentation quality, chemical composition and microbial composition in TMR silages at different moisture levels and different temperatures were also analyzed separately. The results showed that the loss of vitamin A in TMR under different fermentation conditions differed significantly ($p < 0.05$). The increase in moisture level and temperature led to an increase in vitamin A loss, with the greatest loss of up to 68.5% when the moisture level of TMR was 600 g/kg and the storage temperature was 40 °C. In addition, there was a significant correlation ($p < 0.05$) between lactic acid bacteria (LAB) and aerobic bacteria and vitamin A content throughout the fermentation process, thus LAB and aerobic bacteria were associated with vitamin A loss. Thus, a coupling effect of LAB and aerobic bacteria lead to the loss of vitamin A in TMR silage under acidic conditions.

**Keywords:** fermentation quality; moisture level; temperature; total mixed ration silage; vitamin A

## 1. Introduction

Vitamin A is a crucial nutrient that plays a key role in maintaining good health in both humans and animals [1]. In livestock production, it is essential for growth, reproduction, and overall immunity [2]. However, vitamin A is also a highly sensitive nutrient that can be easily degraded under certain storage conditions, including high temperatures and moisture levels [3]. This can have significant implications for the quality and safety of animal feed, particularly in the form of total mixed ration (TMR) silage.

TMR silage is a type of feed commonly used in livestock production, which involves mixing and preserving different types of forages, grains, and supplements to provide a complete and balanced diet for animals [4]. However, the preservation process can be challenging due to various factors that can affect the quality and nutrient content of the silage, including moisture, pH, temperature and oxygen levels. Some studies have described that the nutrient composition of TMR silage, such as non-fibrous carbohydrates [5], neutral detergent fiber (NDF) [6], starch [7] and vitamins [8], is subject to varying degrees of loss during storage. The loss of nutrients undoubtedly increased the waste of energy and the cost of feeding. In order to find the reasons of the nutrient loss of TMR silage and prevent its loss, Hao et al. [9] and Ning et al. [10] studied the protein, starch and hemicellulose loss mechanisms of TMR silages, indicating that microbial enzymes are the main reason for its loss. In addition, Tian et al. [8] found that the vitamin A of TMR lost more than 50% in content after 56 days, but the reason for this loss was not revealed clearly in the article. Among these, temperature and moisture levels have been identified

as key factors that can impact the stability and bioavailability of nutrients in TMR silage [11–13].

Several studies have examined the effects of storage conditions on the change of vitamin A, with varying results depending on the specific conditions and methods used. For instance, Gaylord et al. [14] found that vitamin A of non-fat milk degraded 70% after 48 h of exposure to fluorescence. Moreover, vitamin A is also susceptible to pyrolysis. Suyama et al. [15] investigated the degradation process and degradation products of vitamin A under different heating temperatures (80 °C, 90 °C, 100 °C) and different treatment times (0 min, 15 min, 30 min) in the absence of light. The results showed that the degradation of vitamin A showed positive correlation with temperature and time. When vitamin A palmitate was treated with temperature of 80 °C for 15 min, the degradation rate was found to be about 60%, and when the temperature was increased to 100 °C and the time was extended to 30 min, vitamin A was completely degraded. Soledad et al. [16] reported that vitamin A decreased in liquid infant milk at all storage temperatures (20 °C, 30 °C and 37 °C). In addition, Concepcion et al. [17] found that increasing the activity of water and temperature of storage significantly lowered the retinol content in milk powder. Christian [18] compared the effect of different temperatures and moisture contents on the stability of vitamin A. By measuring the preservation rate of vitamin A, the results showed that vitamin A was preserved to a greater extent at lower temperatures and moisture contents, with, with a preservation rate of 88%, but at higher temperatures and moisture contents, the preservation rate of vitamin A was only 2%. However, there is a considerable lack of information on the possible effects of moisture level and temperature on vitamin A content of TMR silage, especially considering the interaction of temperature and moisture levels on vitamin A stability in TMR silage is not well understood, and further research is needed to identify the optimal conditions for preserving vitamin A in this feed.

To fill this knowledge gap, this study hypothesized that different storage conditions could result in different amounts of vitamin A loss. TMR were stored at different moisture levels (450 g/kg, 525 g/kg and 600 g/kg) and temperatures (15 °C, 30 °C and 40 °C) for 56 days to monitor dynamic changes in vitamin A levels. This article further explained the reasons for the loss of vitamin A in TMR silage under different storage conditions.

## 2. Material and Methods

### 2.1. TMR Preparation

TMR was formulated with alfalfa hay, soybean curd residue, corn meal, soybean meal, vitamin-mineral supplement and salt in a ratio of 36:21:32.5:5:5:0.5, on a basis of dry matter (DM). The moisture levels of TMR were adjusted to 450 g/kg, 525 g/kg and 600 g/kg by adding an appropriate amount of sterile distilled water, respectively. Every 300 g TMR was put into a plastic bag, vacuumed and sealed by a vacuum packer. A total of 9 bags (3 moisture level * 3 replications) from each moisture level were stored at 15 °C, 30 °C and 40 °C incubators, respectively. After 0, 7, 14, 28, 56 days of ensiling, TMR silages were sampled for subsequent analysis, respectively.

### 2.2. Chemical Analysis

The DM was determined by drying the sample at 65 °C for 48 h according to the method 930.15 from the Association of Official Analytical Chemists [19]. Following which, the dried samples were ground to pass a 1-mm screen with laboratory knife mills (FW100, Taisite Instrument Co., Ltd., Tianjin, China). The obtained powder was used to determine water-soluble carbohydrates (WSC), crude protein (CP), neutral detergent fibre (NDF) and acid detergent fibre (ADF) content. The WSC content was measured by the anthrone method [20]. The CP content was analyzed using the methods in AOAC [19]. The aNDF (assayed with alpha-amylase and sodium sulfite) and ADF contents were determined according to the method of Van Soest et al. [21].

Fermentation products of the TMR were determined using water-extracts. Wet TMR (10 g) was homogenized with 90 mL of sterilized distilled water, kept under constant stirring in a homogenizing apparatus (Stomacher 400, Seward, London, UK) for 4 min. The extract was filtered, and the filtrate was used for measuring pH value, ammonia-N and organic acids content. The pH was measured by a glass electrode pH meter (Mettler Toledo S20, Greifensee, Switzerland). The ammonia-N content was analyzed by the method of Broderick and Kang [22]. The organic acids (lactic acid, acetic acid, propionic acid and butyric acid) were determined by high performance liquid chromatography (HPLC, LC-10A, Shimadzu, Tokyo, Japan) as described by Niu et al. [23].

### 2.3. Microbial Analysis

Serial dilutions ($10^{-1}$–$10^{-7}$) were prepared using above water-extract to count for lactic acid bacteria (LAB), aerobic bacteria and yeasts. For LAB counts, de Man, Rogosa and Sharpe agar media (Difco Laboratories, Detroit, MI, USA) were used and incubated at 37 °C for 48 h under anaerobic conditions. Aerobic bacteria were counted on a nutrient agar medium (Nissui, Tokyo, Japan) after incubating at 30 °C for 24 h. For yeast counts, the used culture medium was potato dextrose agar (Nissui, Tokyo, Japan). The incubation was carried out at 30 °C for 48 h, when counting occurred.

### 2.4. Vitamin Analysis

The Vitamin A was determined by the method described by Tian et al. [8]. Briefly, the samples extracted from the TMR silages process were firstly frozen dried by a vacuum freeze-drying machine (FreeZone 4.5 L, LABCONCO Corp., Kansas City, MO, USA), and ground to 1 mm via a mill. Then, the treated samples were saponified and extracted according to the methods of GB/T 17817-2010. Finally, the extracted Vitamin A was analyzed by high performance liquid chromatography (HPLC) equipped with a Pursuit 5 µm C18 column (250 × 4.6 mm; Agilent, Santa Clara, California, USA) and a UV detector for content analysis. The specific detection procedure was as follows: the ratio of methanol to water was 95:5 (*v:v*) as the mobile phase, the flow rate was 1.0 mL/min, and the operating wavelength of the UV detector was 326 nm.

### 2.5. Statistical Analysis

Analysis of variance was performed using the SAS General Linear Models procedure (SAS Institute 9.1 for Windows, SAS Inc., Cary, NC, USA) to determine the general effects of the moisture level, 450 g/kg (M1), 525 g/kg (M2), 600 g/kg (M3), temperature (15 °C, 30 °C, 40 °C) and the interaction between the moisture level and temperature being tested. The treatment means were tested with Tukey's multiple comparisons test. The results for which $p < 0.05$ were regarded as representing a statistically significant difference. The following model was used: $y_{ij} = \mu + \alpha_i + \beta_j + e_{ij}$, where $y_{ij}$ is the observation i and j, $\mu$ is the mean value, $\alpha_i$ is the effect of the moisture level, $\beta_j$ is the effect of the temperature, and $e_{ij}$ are the residuals; correlation analysis between vitamin A and all of the analyzed or measured variables was performed using the SAS Correlation Procedure (SAS Institute 9.1 for Windows, SAS Inc., USA). The Pearson correlation coefficient ($\varrho$) was calculated according to Milton (1992) as $\varrho = \dfrac{\text{Cov}(X,Y)}{\sqrt{(\text{Var } X)(\text{Var } Y)}}$.

## 3. Results

### 3.1. TMR before Ensiling

Table 1 presents the chemical and microbial compositions of TMR mixtures with different moisture levels before ensiling. DM content was affected ($p < 0.001$) by moisture level. The actual moisture level is close to the set value we want to achieve, which is 452 g/kg, 525 g/kg and 577 g/kg, respectively. The pH increased with the increasing moisture

level ($p = 0.047$). There was no significant difference in other chemical composition, microbial composition and vitamins contents among the three moisture levels.

**Table 1.** Chemical and microbial compositions of TMR mixtures with different moisture level before ensiling.

| Items | Moisture Level | | | SEM | *p*-Value |
|---|---|---|---|---|---|
| | **M1** | **M2** | **M3** | | |
| Chemical composition (g/kg DM) | | | | | |
| pH | 6.48 [b] | 6.61 [ab] | 6.69 [a] | 0.039 | 0.047 |
| DM (g/kg FW) | 548 [a] | 475 [b] | 423 [c] | 18.200 | <0.001 |
| CP | 183 | 183 | 178 | 1.600 | 0.278 |
| WSC | 154 | 136 | 143 | 0.390 | 0.182 |
| NDF | 267 | 264 | 260 | 2.575 | 0.617 |
| ADF | 176 | 173 | 173 | 0.837 | 0.129 |
| Microbial composition (log$_{10}$ cfu/g FW) | | | | | |
| LAB | 5.9 | 5.4 | 5.2 | 0.146 | 0.154 |
| Aerobic bacteria | 5.4 | 5.0 | 4.4 | 0.215 | 0.097 |
| Yeasts | 5.9 | 5.8 | 6.0 | 0.104 | 0.884 |
| Vitamin content (mg/kg DM) | | | | | |
| Vitamin A | 3.77 | 3.75 | 3.88 | 0.032 | 0.188 |

DM, dry matter; FW, fresh weight; WSC, water-soluble carbohydrates; CP, crude protein; NDF, neutral detergent fiber; ADF, acid detergent fiber; LAB, lactic acid bacteria. M1, 450 g/kg; M2, 525 g/kg; M3, 600 g/kg; SEM, standard error of the mean; [a–c] Means in the same row with different superscripts differed ($p < 0.05$).

### 3.2. Fermentation Quality and Chemical Composition of TMR Silage

As shown in Table 2, the temperature, moisture and the interaction between them influenced the pH, lactic acid, acetic acid and ammonia-N content ($p < 0.05$). The LAB was affected by both temperature and moisture, whereas aerobic bacteria was only affected by temperature. After 56 days of ensiling, pH decreased while lactic acid, acetic acid and ammonia-N increased with the increase of moisture. Likewise, lactic acid and ammonia-N were also increased with the increase of temperature. In contrast, acetic acid, LAB and aerobic bacteria decreased with the increase of temperature. The yeast count was below the detection limit.

The DM and CP were affected by the moisture level ($p < 0.001$; $p = 0.0013$) (Table 3). The DM, WSC and CP contents decreased with the increase of moisture. However, the WSC content of TMR silage in 30 °C was lowest. There were no effect of moisture and temperature on NDF and ADF contents.

**Table 2.** Fermentation quality and microbial composition of TMR silage after ensiling of 56 days.

| Items | Moisture | Temperature | | | Moisture Mean | SEM | *p*-Value | | |
|---|---|---|---|---|---|---|---|---|---|
| | | **15 °C** | **30 °C** | **40 °C** | | | **T** | **M** | **TxM** |
| pH | M1 | 4.22 | 4.05 | 4.19 | 4.15 [a] | 0.022 | <0.001 | <0.001 | 0.042 |
| | M2 | 4.07 | 3.87 | 4.13 | 4.02 [b] | | | | |
| | M3 | 4.05 | 3.92 | 4.03 | 4.00 [b] | | | | |
| | Temperature mean | 4.11 [x] | 3.95 [y] | 4.12 [x] | | | | | |
| Lactic acid (g/kg DM) | M1 | 62.4 | 74.8 | 64.5 | 67.2 [b] | 0.199 | <0.001 | <0.001 | <0.001 |
| | M2 | 65.1 | 89.8 | 82.0 | 79.0 [a] | | | | |
| | M3 | 65.1 | 83.9 | 83.6 | 77.5 [a] | | | | |

| | | Temperature 15 °C | 30 °C | 40 °C | Moisture Mean | SEM | T | M | TxM |
|---|---|---|---|---|---|---|---|---|---|
| | Temperature mean | 64.2 z | 82.8 x | 76.7 y | | | | | |
| Acetic acid (g/kg DM) | M1 | 4.7 | 4.8 | 4.8 | 4.8 c | 0.019 | 0.002 | <0.001 | 0.023 |
| | M2 | 5.9 | 6.3 | 5.6 | 5.9 b | | | | |
| | M3 | 7.3 | 7.2 | 6.1 | 6.9 a | | | | |
| | Temperature mean | 6.0 x | 6.1 x | 5.5 y | | | | | |
| Ammonia-N (% of total N) | M1 | 1.88 | 1.92 | 2.22 | 2.01 c | 0.104 | <0.001 | <0.001 | 0.010 |
| | M2 | 2.02 | 2.42 | 2.43 | 2.29 b | | | | |
| | M3 | 2.59 | 3.50 | 3.06 | 3.05 a | | | | |
| | Temperature mean | 2.16 y | 2.61 x | 2.57 x | | | | | |
| LAB (log$_{10}$ cfu/g FW) | M1 | 8.6 | 6.5 | 6.9 | 7.3 ab | 0.157 | <0.001 | 0.048 | 0.174 |
| | M2 | 8.0 | 6.5 | 6.7 | 7.1 b | | | | |
| | M3 | 8.2 | 7.1 | 7.1 | 7.5 a | | | | |
| | Temperature mean | 8.3 x | 6.7 y | 6.9 y | | | | | |
| Aerobic bacteria (log$_{10}$ cfu/g FW) | M1 | 3.1 | 3.0 | 2.4 | 2.8 | 0.311 | 0.033 | 0.077 | 0.291 |
| | M2 | 2.6 | 2.6 | 2.4 | 2.5 | | | | |
| | M3 | 2.5 | 2.7 | 2.5 | 2.6 | | | | |
| | Temperature mean | 2.7 x | 2.8 x | 2.4 y | | | | | |
| Yeast (log$_{10}$ cfu/g FW) | M1 | ND | ND | ND | - | - | - | - | - |
| | M2 | ND | ND | ND | - | | | | |
| | M3 | ND | ND | ND | - | | | | |
| | Temperature mean | - | - | - | - | | | | |

DM, dry matter; FW, fresh weight; LAB, lactic acid bacteria. T, Temperature; M, Moisture; TXM, interaction between temperature and moisture; M1, 450 g/kg; M2, 525 g/kg; M3, 600 g/kg; SEM, standard error of the mean; x–z Means in the same row with different superscripts differed ($p < 0.05$); a–c Means in the same column with different superscripts differed ($p < 0.05$).

**Table 3.** Chemical composition of TMR silages.

| Items | Moisture | Temperature 15 °C | 30 °C | 40 °C | Moisture Mean | SEM | *p*-Value T | M | TxM |
|---|---|---|---|---|---|---|---|---|---|
| DM (g/kg FW) | M1 | 539 | 536 | 536 | 537 a | 10.162 | 0.512 | <0.001 | 0.962 |
| | M2 | 464 | 462 | 464 | 463 b | | | | |
| | M3 | 411 | 410 | 412 | 411 c | | | | |
| | Temperature mean | 471 | 469 | 471 | | | | | |
| WSC (g/kg DM) | M1 | 46.4 | 45.1 | 48.5 | 46.7 a | 0.130 | 0.003 | <0.001 | 0.031 |
| | M2 | 47.4 | 32.8 | 41.2 | 40.5 b | | | | |
| | M3 | 35.2 | 33.3 | 40.6 | 36.4 b | | | | |
| | Temperature mean | 43.0 x | 37.0 y | 43.5 x | | | | | |
| CP (g/kg DM) | M1 | 191 | 188 | 187 | 189 a | 0.783 | 0.148 | 0.013 | 0.963 |
| | M2 | 187 | 185 | 184 | 185 ab | | | | |
| | M3 | 184 | 184 | 181 | 183 b | | | | |
| | Temperature mean | 187 | 186 | 184 | | | | | |
| NDF (g/kg DM) | M1 | 268 | 274 | 267 | 270 | 1.832 | 0.939 | 0.993 | 0.892 |
| | M2 | 270 | 268 | 272 | 270 | | | | |
| | M3 | 269 | 268 | 273 | 270 | | | | |

| | | | | | | | | | |
|---|---|---|---|---|---|---|---|---|---|
| | Temperature mean | 269 | 270 | 271 | | | | | |
| ADF (g/kg DM) | M1 | 191 | 191 | 196 | 193 | 0.814 | 0.600 | 0.988 | 0825 |
| | M2 | 193 | 192 | 194 | 193 | | | | |
| | M3 | 194 | 162 | 159 | 193 | | | | |
| | Temperature mean | 193 | 192 | 192 | | | | | |

DM, dry matter; WSC, water-soluble carbohydrates; CP, crude protein; NDF, neutral detergent fiber; ADF, acid detergent fiber; T, Temperature; M, Moisture; TXM, interaction between temperature and moisture; M1, 450 g/kg; M2, 525 g/kg; M3, 600 g/kg; SEM, standard error of the mean; [x,y] Means in the same row with different superscripts differed ($p < 0.05$); [a–c] Means in the same column with different superscripts differed ($p < 0.05$).

### 3.3. Change of Microbial Composition in TMR Silage

Figure 1 shows changes in the number of microbials in TMR silage. In different moisture levels, LAB, aerobic bacteria counts increased rapidly at the first 7 days of ensiling, whereas yeast decreased rapidly. After 7 days, LAB and yeast counts showed a downward trend, while aerobic bacteria showed a sharp decrease. Likewise, the changes in the count of microbials under different temperature conditions showed similar trends. However, it could be observed that temperature has a greater effect on microbial counts when compared to the effect of moisture. Throughout the fermentation process, the number of LAB and yeast were consistently higher at 15 °C than at 30 °C and 40 °C (Figure 1b,f). However, the number of aerobic bacteria at 40 °C was consistently higher than that at 15 °C and 30 °C during the first 14 days of ensiling. After 14 days of ensiling, their population levels were comparable. Therefore, high temperature had an inhibitory effect on LAB and yeast, but it is conducive to the reproduction of aerobic bacteria.

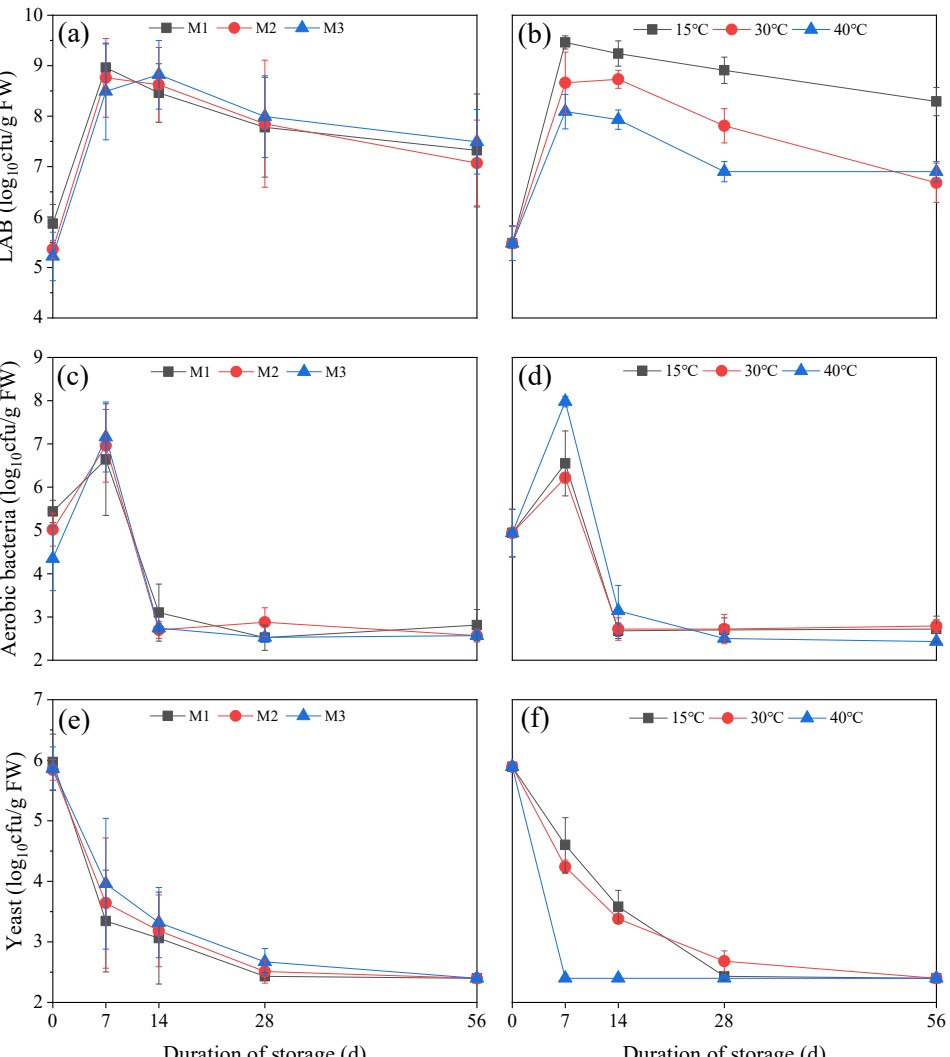

**Figure 1.** (**a**,**c**,**e**) were dynamic changes in LAB, aerobic bacteria and yeast number of TMR silage stored under different moisture levels. (**b**,**d**,**f**) were dynamic changes in LAB, aerobic bacteria and yeast number of TMR silage stored under different temperature conditions. LAB, lactic acid bacteria; M1, 450 g/kg; M2, 525 g/kg; M3, 600 g/kg; Data are presented as means of three replicates (Mean ± Standard error).

### 3.4. Vitamins' Changes of TMR Silage

Vitamin A of the TMR silages during 0 to 56 days of storage are shown in Figure 2. The vitamin A content decreased rapidly on days 0 to 7, however decreased slowly after the 7th day. After 56 days of ensiling, the vitamin A loss reached the range of 45.9–68.5% (Figure 3). It was observed that moisture and temperature had significant effects on the vitamin A loss. The vitamin A loss of M2 and M3 increased with increasing temperature ($p < 0.05$). There was no significant difference of vitamin A loss between 30 °C and 40 °C at the M1 moisture level. However, the vitamin A losses were statistically significantly higher in both 30 °C and 40 °C than at 15 °C, at the M1 moisture level. Further, with the increase in moisture level, the vitamin A loss increased at 15 °C ($p < 0.05$). The vitamin A losses were not significantly different in the moisture levels M2 and M3 when TMR was stored at both 30 °C and 40 °C conditions, but both were significantly higher than the M1 moisture level.

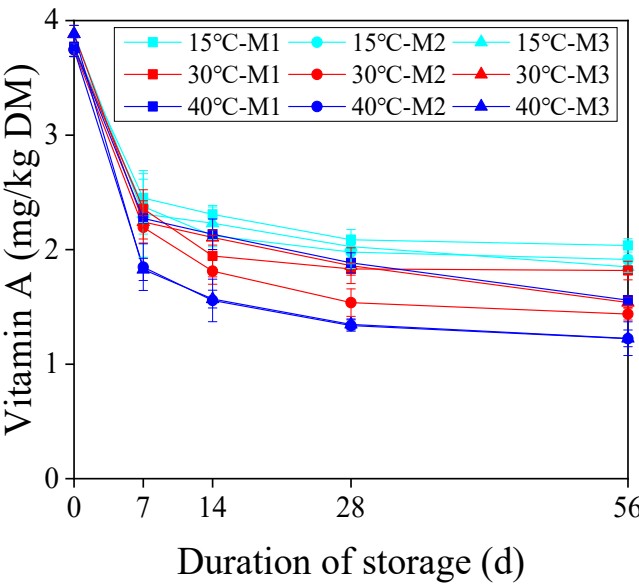

**Figure 2.** Changes of vitamin A and vitamin E contents with the duration of storage under different storage conditions. M1, 450 g/kg; M2, 525 g/kg; M3, 600 g/kg. Data are presented as the means of three replicates (Mean ± Standard error).

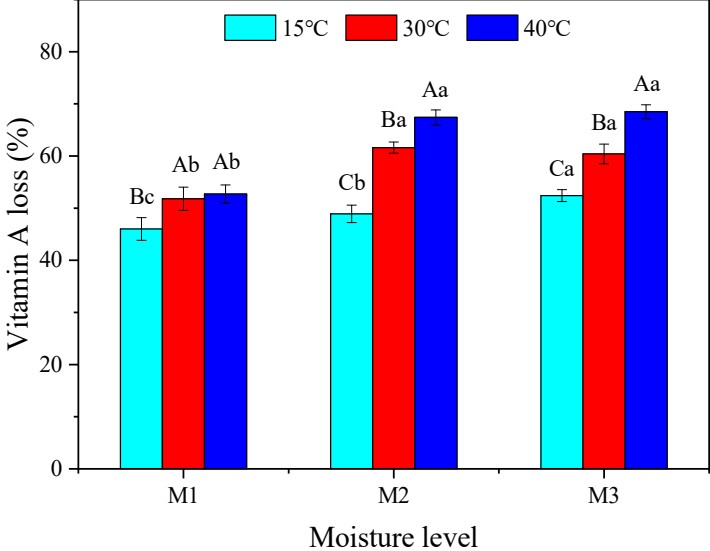

**Figure 3.** Vitamin A loss with the temperature at different moisture levels. M1, 450 g/kg; M2, 525 g/kg; M3, 600 g/kg. Lowercase letters represent the level of significant difference between samples stored at different moisture levels for the same temperature ($p < 0.05$). Capital letters represent the level of significant difference between samples stored at the same moisture levels for different temperatures ($p < 0.05$). ($n = 3$, bars indicate the standard error of the means).

### *3.5. Correlation between Vitamin A and Other Various Variables*

As shown in Table 4, in the early fermentation stage (0–7 days) of TMR with different moisture levels, the vitamin A content was significantly positively correlated with WSC, pH and yeast ($p < 0.001$), and negatively correlated with lactic acid, acetic acid, LAB and aerobic bacteria ($p < 0.001$). In the later fermentation stage (14–56 days), the correlation between vitamin A and WSC, pH, lactic acid, acetic acid, aerobic bacteria and yeast weakened or disappeared. With the increase of moisture level, the correlation between LAB and vitamin A weakened in the early stage but increased in the later stage. However, the correlation between aerobic bacteria and vitamin A increased in the early stage. The correlation between yeast and vitamin A was weakened in both early and later stages.

**Table 4.** Correlation between vitamin A and other measured or analyzed variables (TMR silage with different moisture level).

| Variable | M1-Vitamin A (mg/kg DM) | | M2-Vitamin A (mg/kg DM) | | M3-Vitamin A (mg/kg DM) | |
|---|---|---|---|---|---|---|
| | 0–7 day | 14–56 day | 0–7 day | 14–56 day | 0–7 day | 14–56 day |
| DM (g/kg FW) | 0.334 | 0.545 | 0.350 | −0.079 | 0.338 | 0.039 |
| CP (g/kg DM) | 0.109 | −0.445 | 0.002 | 0.228 | 0.025 | −0.126 |
| NDF (g/kg DM) | −0.460 | −0.396 | −0.491 | 0.136 | −0.440 | −0.252 |
| ADF (g/kg DM) | −0.286 | −0.165 | −0.257 | −0.198 | −0.287 | −0.274 |
| WSC (g/kg DM) | 0.999 ** | 0.751 * | 0.970 ** | 0.651 | 0.997 ** | 0.465 |
| pH | 0.998 ** | 0.790 * | 0.988 ** | 0.740 * | 0.988 ** | 0.711 * |
| Lactic acid (g/kg DM) | −0.988 ** | −0.826 ** | −0.996 ** | −0.769 * | −0.986 ** | −0.748 * |
| Acetic acid (g/kg DM) | −0.998 ** | −0.426 | −0.981 ** | −0.174 | −0.952 ** | 0.617 |
| LAB ($\log_{10}$ cfu/g FW) | −0.970 ** | 0.740 * | −0.901 * | 0.920 ** | −0.899 * | 0.943 ** |
| Aerobic bacteria ($\log_{10}$ cfu/g FW) | −0.878 ** | 0.309 | −0.947 ** | 0.274 | −0.976 ** | 0.500 |
| Yeasts ($\log_{10}$ cfu/g FW) | 0.969 ** | 0.764 * | 0.945 ** | 0.528 | 0.803 * | 0.524 |

* 95% confidence level; ** 99% confidence level; DM, dry matter; WSC, water-soluble carbohydrates; CP, crude protein; NDF, neutral detergent fiber; ADF, acid detergent fiber; LAB, lactic acid bacteria.

As shown in Table 5, the vitamin A content of TMR silage stored at a different temperature was significantly positively correlated with WSC, pH and yeast ($p < 0.001$), but negatively correlated with lactic acid, acetic acid, LAB and aerobic bacteria in the early stage (0–7 days) ($p < 0.001$). However, in the later fermentation stage (14–56 days), the correlation between vitamin A content and pH, lactic acid, LAB and yeast weakened, and the correlation with WSC, acetic acid and aerobic bacteria disappeared. In the early fermentation stage (0–7 days), as the temperature increases, the correlation between aerobic bacteria and vitamin A increases, and yeast has the strongest correlation with vitamin A at 30 °C. In the later fermentation stage (14–56 days), only aerobic bacteria at 40 °C were correlated with vitamin A, and the correlation between yeast and vitamin A disappeared.

**Table 5.** Correlation between vitamin A and other measured or analyzed variables (TMR silage stored at different temperature).

| Variable | 15 °C-Vitamin A (mg/kg DM) | | 30 °C-Vitamin A (mg/kg DM) | | 40 °C-Vitamin A (mg/kg DM) | |
|---|---|---|---|---|---|---|
| | 0–7 day | 14–56 day | 0–7 day | 14–56 day | 0–7 day | 14–56 day |
| DM (g/kg FW) | 0.026 | 0.363 | 0.096 | 0.131 | 0.088 | 0.350 |
| CP (g/kg DM) | 0.512 | −0.500 | −0.023 | 0.081 | −0.372 | 0.441 |
| NDF (g/kg DM) | −0.578 | −0.423 | −0.537 | 0.304 | −0.550 | −0.166 |
| ADF (g/kg DM) | −0.309 | −0.472 | −0.429 | −0.441 | −0.374 | −0.407 |
| WSC (g/kg DM) | 0.984 ** | 0.199 | 0.988 ** | 0.184 | 0.984 ** | 0.269 |
| pH | 0.999 ** | 0.939 ** | 0.998 ** | 0.859 ** | 0.993 ** | 0.918 ** |
| Lactic acid (g/kg DM) | −0.994 ** | −0.941 ** | −0.996 ** | −0.868 ** | −0.996 ** | −0.953 ** |
| Acetic acid (g/kg DM) | −0.956 ** | −0.525 | −0.960 ** | −0.221 | −0.944 ** | 0.333 |
| LAB ($\log_{10}$ cfu/g FW) | −0.996 ** | 0.729 * | −0.967 ** | 0.681 * | −0.945 ** | 0.669 * |
| Aerobic bacteria ($\log_{10}$ cfu/g FW) | −0.871 * | −0.013 | −0.915 * | −0.201 | −0.966 ** | 0.750 * |
| Yeasts ($\log_{10}$ cfu/g FW) | 0.945 ** | 0.868 ** | 0.990 ** | 0.686 * | 0.987 ** | - |

* 95% confidence level; ** 99% confidence level; DM, dry matter; WSC, water-soluble carbohydrates; CP, crude protein; NDF, neutral detergent fiber; ADF, acid detergent fiber; LAB, lactic acid bacteria.

## 4. Discussion

### 4.1. Fermentation Quality of TMR Silage

pH is generally used to monitor the silage quality, and the pH range of 3.8–4.2 is desired for a successful ensiling [24]. Moreover, He et al. [25] reported that pH 4.2 is generally regarded as a benchmark for well-preserved silage, especially for high moisture silages. In this study, pH values of almost all TMR decreased to below 4.2 after 56 days of ensiling, except for the TMR of the M1 moisture level stored at 15 °C (Table 2). Moreover, the count of LAB is high at 15 °C (Figure 1b). This result can be explained by the temperature influencing the ecophysiological properties of LAB. The lower moisture and temperature restricted silage fermentation [26,27]. However, other TMR silages had a good fermentation quality, as indicated by the large increase in lactic acid concentration and decrease in pH value, with small increases in acetic acid and ammonia-N concentrations due to the ensiling process. In particularly, the TMR of the M2 moisture level stored at 30 °C had the best fermentation quality, with the highest lactic acid content (82.8 g/kg DM) and the lowest pH value (3.87). This is not consistent with the findings of Ali et al. [28] who reported that silage could be better preserved at 40 °C than 30 °C. This difference may be attributed to the variation of moisture level. Besides, Li et al. [29] and Liu et al. [30] indicated that high temperature weakened the LAB homo-fermentation so that the TMR silage of 30 °C a had higher lactic acid content than that of 40 °C in this study.

As another important indicator, an ammonia-N content of lower than 5% TN is desired as an excellent TMR silage. All TMR silage in this experiment met this standard, and the increases of moisture level and temperature had improved the ammonia-N content. Similar results were found in Hao et al. [26] and Makoto et al. [31] who indicate that lower moisture levels and temperatures suppressed the deamination within TMR silage.

### 4.2. Vitamin A Change of TMR Silage

In our previous study, vitamin A changes of TMR ensiled by different type of herbage were investigated [8]. The results showed that the loss of vitamin A in the early stage of fermentation (0–7 day) was much higher than that in the later stage of fermentation (14–56 day). Through analysis of the data, it was concluded that this phenomenon is mainly due to the production of a large amount of lactic acid in the early stage of fermentation, which leads to a significant decrease in pH. Moreover, Erdman et al. [32] reported that vitamin A content will be lost when the pH is below 4.5. However, it is interesting in our previous study that the pH values were above 4.5 in the early fermentation stage. Therefore, we speculate that vitamin A maybe also be unstable at pH > 4.5. Moreover, there must be other factors that accelerate the losses of vitamin A in the early stage except for the acid condition.

In the present study, it can be observed that different moisture level and temperature have significant effects on the loss of vitamin A in TMR silage. Under the moisture levels of M2 and M3, the vitamin A loss of the TMR increased significantly with the increase of temperature, which indicated that high temperature was not conducive to the preservation of vitamin A. However, there was no significant difference in vitamin A loss between 30 °C and 40 °C in TMR at an M1 moisture level, but both were higher than 15 °C. In addition, at the same temperature, the loss of vitamin A in M2 and M3 was significantly higher than that in M1. This indicates that the moisture level has a great influence on the loss of vitamin A.

The correlation analysis showed that there was a significant correlation between vitamin A and LAB throughout the fermentation process in TMR silage with different moisture levels. Interestingly, the correlation between vitamin A and LAB weakened as the moisture level increased in the early fermentation stage, whereas this correlation was enhanced in the late fermentation stage. This may due to the higher moisture level contribution to the growth of aerobic bacteria and yeast in the early fermentation stage (Figure 1c,e), which could compete with LAB. Along with the fermentation process, LAB became

dominant and the pH, and aerobic bacteria and yeasts were inhibited. Thus, this could explain that the stronger correlation between vitamin A and LAB in the late fermentation stage. Moreover, the dynamic changes of LAB, aerobic bacteria and yeast, and the correlation changes between vitamin A and aerobic bacteria and yeast can also confirm this point. Although both aerobic bacteria and yeast correlate with vitamin A content in the early stages of fermentation, the correlation weakened or even disappeared in the later stages of fermentation. Therefore, we infered that these two factors may not be the main cause of vitamin A loss. Furthermore, there are no reports on whether aerobic bacteria and yeasts play a role in the loss of vitamin A during the early stages of fermentation. The results of the present study did not elucidate this issue either, therefore further research is needed. However, it can be confirmed that LAB have an impact on the loss of vitamin A since the correlation between LAB and vitamin A was maintained throughout the fermentation process. Panfili et al. [33] reported that some LAB could decrease the potency of vitamin A by causing an isomerization reaction of vitamin A from all-trans to cis isomers. Moreover, the higher moisture level promoted acid production by LAB, which accelerates the loss of vitamin A. In addition, water is a polar solvent, and vitamin A is sensitive to it. Rutkowski et al. [34] studied the vitamin A stability in triple fortified salt with different humidity levels. The results showed that the retention rate of vitamin A at 60% relative humidity was 1.5 times that at 100%. This indicated that relatively high humidity increases the loss of vitamin A.

The effect of temperature on the stability of vitamin A in TMR silage was also investigated in this paper, and the results proved that temperature is another important factor affecting the stability of vitamin A. The increase in temperature led to an increased loss of vitamin A. In particular, the TMR silage at the moisture level of M3 showed the greatest loss (68.5%) at a storage temperature of 40 °C. Frias et al. [35] reported that the loss of vitamin A in commercially available formulas stored at 30 °C in the dark for 3, 6 and 9 months was greater than that at 4° and 20° conditions. Meanwhile, Parrish et al. [36] reported that the loss rate of vitamin A at a room temperature for 1 month was much greater than that when stored in a refrigerator. This is consistent with the results of the present study, where high temperature and high humidity are very unfavorable for the preservation of vitamin A. The correlation analysis of vitamin A of TMR stored at different temperatures with other variables revealed that the results were similar to the correlation analysis of vitamin A of TMR stored at different moisture contents with other variables, and there was a significant correlation between vitamin A and LAB throughout the fermentation process. However, the difference was that the correlation between vitamin A and LAB weakened with increasing temperature in the later stages of fermentation, and even disappeared at 40 °C. This is because later in the fermentation, the lower temperature favors the survival of yeast, thus remaining in competition with LAB.

In addition, the correlations between vitamin A and the aerobic bacteria and yeast at different temperatures also differed. At the late stage of fermentation, aerobic bacteria had stabilized at 15 °C and 30 °C, while aerobic bacteria still slightly decreased at 40 °C. Therefore, there was still a significant correlation between vitamin A and aerobic bacteria at 40 °C. The correlation between yeast and vitamin A diminished or even disappeared as the temperature increased, due to the fact that high temperatures are not conducive to the survival of yeast [37,38]. Despite the higher number of yeast organisms at 15 °C, the rate of vitamin A loss was lower. So perhaps yeast does not play a role in the loss of vitamin A, but this still needs further study.

To this step, we cannot be sure that whether aerobic bacteria and yeast contributed the loss of vitamin A, and if so, how much their contribution is. Further analysis may be directed towards whether the high moisture content is favorable for LAB, aerobic bacteria and yeast to multiply (Figure 1a,c,e). However, high temperature is not favorable to LAB and yeast reproduction, but favorable to aerobic bacteria reproduction (Figure 1b,d,f). In addition, high temperature and moisture level would accelerate the loss of vitamin A. Therefore, we dare to suspect that the coupling of LAB and aerobic bacteria may also have

contributed to the loss of vitamin A. Therefore, based on the above analysis, we think that a combination of acidic conditions, lactic acid bacteria and aerobic bacteria lead to the loss of vitamin A in TMR silage.

## 5. Conclusions

In this study, the best fermentation quality of TMR was obtained when the moisture content was 525 g/kg and the storage temperature was 30 °C. High temperatures and high moisture levels were not conducive to the preservation of vitamin A. In addition, a combination of acidic conditions, LAB and aerobic bacteria leads to the loss of vitamin A in TMR silage. But further research is needed on how LAB and aerobic bacteria affect the loss of vitamin A.

**Author Contributions:** Methodology, P.T., H.H., X.Z. and M.C.; Software, X.Z. and M.C.; Formal analysis, P.T. and H.H.; Investigation, P.T.; Resources, P.T.; Writing—original draft, P.T.; Writing—review and editing, X.W. All authors have read and agreed to the published version of the manuscript.

**Funding:** This research was funded by Xiangyang Science and Technology Plan Project, Hubei, China (2021ABA003618) and the Foundation of Education Commission of Hubei University of Arts and Science (kyqdf2021011 and kyqdf2022007).

**Institutional Review Board Statement:** Not applicable.

**Informed Consent Statement:** Not applicable.

**Data Availability Statement:** Data are available from the authors.

**Conflicts of Interest:** The authors declare no conflict of interest.

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
