# Peer review of "Effects of Temperature and Moisture Levels on Vitamin A in Total Mixed Ration Silage"

_fermentation, doi:10.3390/fermentation9070614_

Round 1

Reviewer 1 Report

This paper reports valuable data on vitamin A loss in TMR silage. Below is a summary of the feedback received.

2. Materials and methods

2.2. Chemical analysis

P2L63: In the preprocessing of the samples, when analyzing samples containing volatile components such as organic acids, it is desirable to use freeze-drying. Why was the drying method of 65°C for 48 hours chosen?

3. Results

3.5. Correlation between vitamin A and other various variables

P7L181 Investigating the relationship between each component value and the number of microorganisms and vitamin A for each fermentation stage. However, the method does not explain how to study each fermentation step. Regarding this result, how was the sample of the fermentation process done?

4. Discassion

4.1 Fermentation quality of TMR silage

The relationship between lactic acid content and LAB is described, but I think it would be better to organize it a little more. It has been reported that TMR silage tends to suppress the activities of yeast and the like more than silage preparation of a single raw material such as pasture grass. For this reason, I think it is better to organize and describe the requirements related to bacterial activity. First, with regard to moisture content, the activity of bacteria involved in silage fermentation should be suppressed as the moisture content decreases. I think that it is necessary to consider water activity. Next, how about considering the optimum temperature of the LAB? Judging from the pH and lactic acid content in this paper, 30°C  is considered to be good fermentation in all test plots.

Since the discussion in this section is a precondition for the discussion in 4.2. a thorough explanation may be necessary.

4.2. Vitamin A change of TMR silage

First, the common depletion factors of vitamin A should be organized and described. Next, if you consider how the results obtained in this experiment are connected, I think you can organize various factors such as vitamin A and other component values and bacteria. In addition, if we organize the relationship between the activity of bacteria and the amount of organic acid produced as a result, we will naturally be able to see the effect on the loss of micronutrients such as vitamins.

5. Conclusion

In 4.2., is the loss of vitamin A affected by LAB? From the results of this paper, it seems that the relationship with the content of organic acids such as lactic acid is higher than that of LAB, so it is natural to investigate the relationship between acidic substances in fungi and draw a conclusion.

Author Response

Reviewer 1:

This paper reports valuable data on vitamin A loss in TMR silage. Below is a summary of the feedback received.

  1. Materials and methods

2.2. Chemical analysis

P2L63: In the preprocessing of the samples, when analyzing samples containing volatile components such as organic acids, it is desirable to use freeze-drying. Why was the drying method of 65°C for 48 hours chosen?

Response: I apologies for that. The drying method was only chosen for determining the DM, WSC, CP, NDF and ADF. Likewise, the volatile components were using water-extracts. Please see P3L97.

  1. Results

3.5. Correlation between vitamin A and other various variables

P7L181 Investigating the relationship between each component value and the number of microorganisms and vitamin A for each fermentation stage. However, the method does not explain how to study each fermentation step. Regarding this result, how was the sample of the fermentation process done?

Response: Thanks for your valuable comments. I provide a detailed description of statistical analysis. Please see 2.5 Statistical analysis [P3L125-137].

  1. Discassion

4.1 Fermentation quality of TMR silage

The relationship between lactic acid content and LAB is described, but I think it would be better to organize it a little more. It has been reported that TMR silage tends to suppress the activities of yeast and the like more than silage preparation of a single raw material such as pasture grass. For this reason, I think it is better to organize and describe the requirements related to bacterial activity. First, with regard to moisture content, the activity of bacteria involved in silage fermentation should be suppressed as the moisture content decreases. I think that it is necessary to consider water activity. Next, how about considering the optimum temperature of the LAB? Judging from the pH and lactic acid content in this paper, 30°C  is considered to be good fermentation in all test plots.

Since the discussion in this section is a precondition for the discussion in 4.2. a thorough explanation may be necessary.

Response: I agree with your suggestion. In our previous study, we investigated vitamin A changes of TMR ensiled by different type of herbage[1]. The results showed that the loss of vitamin A in the early stage (0-7d) of fermentation was much higher than that in the later stage (14-56d) of fermentation. Through analysis of the data, we think that this phenomenon is mainly due to the production of a large amount of lactic acid in the early stage of fermentation, which leads to a significant decrease in pH. Moreover, Erdman et al. [2] reported that vitamin A content will be lost when the pH is below 4.5. However, it is interesting in our study that the pH values were above 4.5 in the early fermentation stage. Therefore, we speculate that vitamin A maybe also be unstable at pH>4.5. Moreover, there must be other factors that accelerate the losses of vitamin A in the early stage except for the acid condition.

It is known that the early fermentation stage of TMR is a time of dramatic changes in biochemical conditions. LAB proliferate in large numbers, yeasts are inhibited, while aerobic bacteria first increase and then be inhibited(due to the presence of residual oxygen). In the present study, what we can determine is that LAB are correlated with the loss of vitamin A. However, we cannot determine whether aerobic bacteria and yeast contributed the loss of vitamin A, and if so, how much their contribution is. The high moisture content is favorable for LAB, aerobic bacteria and yeast to multiply (Figure 1a,c,e). However, high temperature is not favorable to LAB and yeast reproduction, but favorable to aerobic bacteria reproduction (Figure 1b,d,f). In addition, high temperature and moisture level would accelerate the loss of vitamin A. Therefore, we suspect that the coupling of lactic acid bacteria and aerobic bacteria may have contributed to the loss of vitamin A.

To summarize, we think that a combination of acidic conditions, lactic acid bacteria and aerobic bacteria lead to the loss of vitamin A in TMR silage.

Reference

[1] Tian P., Niu D., Zuo S., Jiang D., Li R., Xu C. Vitamin A and E in the total mixed ration as influenced by ensiling and the type of herbage. Science of The Total Environment, 2020, 746:141239.

[2] Erdman Jr, J., Poor, C., Dietz, J. 1988. Factors affecting the bioavailability of vitamin A, carotenoids, and vitamin E. Food technology (USA).

4.2. Vitamin A change of TMR silage

First, the common depletion factors of vitamin A should be organized and described. Next, if you consider how the results obtained in this experiment are connected, I think you can organize various factors such as vitamin A and other component values and bacteria. In addition, if we organize the relationship between the activity of bacteria and the amount of organic acid produced as a result, we will naturally be able to see the effect on the loss of micronutrients such as vitamins.

Response: Bundle of thanks for the suggestion. The following text is incorporated in the revised manuscript [P9L271-280].

  1. Conclusion

In 4.2., is the loss of vitamin A affected by LAB? From the results of this paper, it seems that the relationship with the content of organic acids such as lactic acid is higher than that of LAB, so it is natural to investigate the relationship between acidic substances in fungi and draw a conclusion.

Response: Thanks for the suggestion. To summarize, we think that a combination of acidic conditions, lactic acid bacteria and aerobic bacteria lead to the loss of vitamin A in TMR silage. The revised text is P11L356-360

Reviewer 2 Report

General consideration: The aim of this study was to evaluate the effects of temperature and moisture levels on vitamin A in the total mixed ration (TMR) silage.

The introduction should be improved by reporting more references on the effect of temperature and humidity levels on the vitamin A content of TMR administered to animals. The materials and methods section lacks information which is then discussed in the results section.

Weak conclusions. No possible spin-offs are discussed. I'm just describing a phenomenon.

 2. Materials and Methods

2.1 TMR preparation

There is no information on the chemical composition of the feed used for the preparation.

There is no information on how the humidity levels were regulated.

There is no information on how they were ensiled, such as whether bacteria were added or otherwise.

The storage temperatures reported, especially the second and third, differ by 5°C from those discussed below.

 The paragraph relating to statistical processing is missing

 3. Results

3.1 e 3.2

too synthetic

3.3. Change of microbial composition in TMR silage

L. 153-154 Hypothesis or statement? In the second case, support better

 4. Discussion

 4.2 Vitamin A change of TMR silage

Add references.

 5. Conclusion

Weak conclusions. No possible spin-offs are discussed. I'm just describing a phenomenon.

Author Response

Reviewer 2:

General consideration: The aim of this study was to evaluate the effects of temperature and moisture levels on vitamin A in the total mixed ration (TMR) silage.

The introduction should be improved by reporting more references on the effect of temperature and humidity levels on the vitamin A content of TMR administered to animals. The materials and methods section lacks information which is then discussed in the results section.

Response: Thanks for the suggestion. The revised text has been added to the introduction [P2L47-66] and materials and methods [P2L79-86]

Weak conclusions. No possible spin-offs are discussed. I'm just describing a phenomenon.

Response: I apologize for that. The following revised text has been added to the conclusions. [P11L356-360]

In this study, the best fermentation quality of TMR was obtained when the moisture content was 525 g/kg and the storage temperature was 30°C. High temperature and high moisture level were not conducive to the preservation of vitamin A. In addition, a combination of acidic conditions, LAB and aerobic bacteria lead to the loss of vitamin A in TMR silage. But further research is needed on how LAB and aerobic bacteria affect the loss of vitamin A.

  1. Materials and Methods

2.1 TMR preparation

There is no information on the chemical composition of the feed used for the preparation.

Response: I apologize for that. This study mainly focuses on the chemical composition of TMR before ensiling. So we did not separately determine the chemical composition of each ingredient.

There is no information on how the humidity levels were regulated.

Response: The moisture levels of TMR were adjusted to 450 g/kg, 525 g/kg and 600g/kg by adding an appropriate amount of sterile distilled water, respectively. Please see P2L82

There is no information on how they were ensiled, such as whether bacteria were added or otherwise.

Response: I added a detailed description about ensiled information in paragraph 2.1. TMR preparation[P2L79-86]. No bacteria or otherwise was added for ensiling.

The storage temperatures reported, especially the second and third, differ by 5°C from those discussed below.

Response: Thank you for pointing out the error. I apologize for that. The right storage temperature is 15℃,30℃ and 40℃. I have corrected the error.

 The paragraph relating to statistical processing is missing

Response: Thanks for pointing out. I provide a detailed description of statistical analysis. Please see P3L125-137

  1. Results

3.1 e 3.2

too synthetic

Response: I apologize for that. 3.1 and 3.2 are mainly the analysis of the chemical composition and fermentation quality of TMR before and after 56 days of ensiling.

3.3. Change of microbial composition in TMR silage

  1. 153-154 Hypothesis or statement? In the second case, support better

Response: It’s a statement. We can obtain this information from Figure 1. The revised text has been added to P6L183-188.

  1. Discussion

4.2 Vitamin A change of TMR silage

Add references.

Response: Thanks for the suggestion. I have added references. Please see P9L271-280

  1. Conclusion

Weak conclusions. No possible spin-offs are discussed. I'm just describing a phenomenon.

Response: Thanks for your valuable comments. The following revised text has been added to the conclusions. [P11L354-360]

In this study, the best fermentation quality of TMR was obtained when the moisture content was 525 g/kg and the storage temperature was 30°C. High temperature and high moisture level were not conducive to the preservation of vitamin A. In addition, a combination of acidic conditions, LAB and aerobic bacteria lead to the loss of vitamin A in TMR silage. But further research is needed on how LAB and aerobic bacteria affect the loss of vitamin A.